# Energy Transfer in Incompressible Magnetohydrodynamics: The Filtered Approach

**Jesse T. Coburn** [1,*] **and Luca Sorriso-Valvo** [2,3] 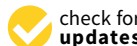

[1]    School of Physics and Astronomy, Queen Mary University of London, London E1 4NS, UK
[2]    Departamento de Física, Escuela Politécnica Nacional, Quito 170517, Ecuador
[3]    Istituto per la Scienza e Tecnologia dei Plasmi–CNR, Sede di Bari, 70126 Bari, Italy
[*]    Correspondence: j.t.coburn@qmul.ac.uk

**Abstract:** We develop incompressible magnetohydrodynamic (IMHD) energy budget equations with a spatial filtering kernel and estimate the scaling of the structure functions. The Politano-Pouquet law is recovered as an upper bound on the scale-to-scale energy transfer. The primary result of this work is the relation of the scaling of IMHD invariants. It can be produced by hypothesizing a scale-independent energy transfer rate. These results have relevance in plasma regimes where the approximations of IMHD are justified. We measure structure functions with solar wind data and find support for the relations.

**Keywords:** magnetohydrodynamics; turbulence

## 1. Introduction

In turbulent flows, obtaining theoretical solutions for the field (velocity, density, magnetic, etc.) structure is made difficult by the problem of closure of the equations (e.g., Navier-Stokes). The standard way to skirt this issue is to propose a self-similar relation on the statistical distribution function. This is the key content of Andrei Kolmogorov's breakthrough in turbulence theory [1]. This connection is born from a result on the Navier-Stokes equation for stationary, statistically homogeneous, ideal turbulence (large Reynolds' number or vanishing viscosity) [2,3]. Kolmogorov noted that a set of scales will be dominated by non-linear interactions that are independent of the viscous details. This represents a definition of universality, which together with the hypothesis of constant skewness of the fluctuations, allows one to theorize the shape of the power spectrum of velocity fluctuations at intermediate scales [4]. A wealth of research followed these efforts introducing corrections and further questions on the precise mathematical account of the scale similarity.

Kolmogorov's phenomenological result can also be produced in a seemingly more rigorous fashion by introducing a filter [5], identifying the scale-to-scale energy flux and using estimates [6]. This approach leads to the consideration that, for ideal turbulence, the energy will not reshuffle, say among the Fourier modes of the velocity field, unless the velocity field suffices a Hölder condition. This relates the increments of the velocity field to its spatial scaling as Kolmogorov had done (although with some differences, which are noted below). In the literature, this is termed the "Onsager minimal assumption" and has been charmingly recapped, among other non-published works, by Lars Onsager in [7]. In this article, we follow the general method put forth by Onsager, but applied to the incompressible magnetohydrodynamic (IMHD) equations instead of Navier–Stokes, and derive a set of scaling exponents. Although this is not a novel result [8,9], we have recapped the derivation with precise detail and an alternative set of scaling exponents, for reasons we briefly expose in the following.

The power spectral density of the interplanetary magnetic field was measured by the Mariner 2 spacecraft and presented by Paul J. Coleman Jr. [10] in a formative article that birthed the research

field of solar wind turbulence. Direct measurement of the solar wind, a hot tenuous rarely collisional ionized gas, has presented physicists a massive test bed for plasma and turbulence theories [11]. Among the many persisting topics with unresolved theory, the prediction of the power spectral density plays a central role. Formalized versions of Kolmogorov's work, called the Kármán-Howarth-Monin relations, have been applied to various versions of the magnetohydrodynamic equations [12], but the scale-similarity assumptions are complicated by an anisotropy to the direction of the magnetic field. Notable attempts on the subject have been made with degrees of success [13–15]. However, thus far, there is no exact theory for the form of the IMHD power spectral density. For this reason, we seek here a different approach.

## 2. Definitions and Notation

Let us begin by defining a general spatial filtering kernel $\Im_\ell(x - x')$ that is positive definite, normalized, centered and rapidly decreasing towards infinity so it satisfies the following properties:

$$\Im_\ell(x - x') \geq 0; \qquad \lim_{r \to \pm \ell} \Im_\ell(x - x') \to 0 \tag{1}$$

$$\int_{V(\ell)} d^3x' \, \Im_\ell(x - x') = 1 \tag{2}$$

$$\int_{V(\ell)} d^3x' \, \Im_\ell(x - x') \, |x_i - x'_i| = 0 \tag{3}$$

$$\Im_\ell(x - x') = \ell^{-3} \, \Im_\ell(U) \, ; \quad U = \frac{|x_i - x'_i|}{\ell} \qquad \lim_{\ell \to 0} \ell^{-3} \, \Im_\ell(|x_i - x'_i|/\ell) \to \Delta(|x_i - x'_i|) \, , \tag{4}$$

where the separation vector $r = x' - x$ is written bold as argument of a function and with index notation outside, and the norm is defined as $|r_i| = \sqrt{r_i r_i}$. The above kernel ensures the standard convolution integral definition. The quantity $\ell$ is the radius of the integration volume, and it is not identified with any physically identifiable scale. This can also be seen in the right-hand limit of Equation (4), where the unorthodox symbol $\Delta$ is used (for notational issues) to indicate the Dirac delta function. The large- and small-scale filters on a generic vector field $y$ are defined, respectively, as:

$$\widetilde{y}_i^\ell(x) = \int_{V(\ell)} d^3x' \, \Im_\ell(x - x')y_i(x') \tag{5}$$

$$\bar{y}_i^\ell(x) = y_i(x) - \widetilde{y}_i^\ell(x) = -\int_{V(\ell)} d^3x' \, \Im_\ell(x - x')\delta y_i(x; r) \, , \tag{6}$$

with the definition $\delta y_i(x; r) = y_i(x + r) - y_i(x)$ indicating the standard two-point, scale-dependent field increment. From Equation (4), it follows that, at small-scales, $\int_{V(\ell)} d^3x' \Im_\ell(x - x')f(x) = f(x) \int_{V(\ell)} d^3x' \Im_\ell(x - x') = 1$.

### 2.1. Commutation with Derivative

To display further properties, we show that the derivative commutes with the filter:

$$\widetilde{\partial_j f(x)}^\ell = \int_{V(\ell)} d^3x' \, \Im_\ell(x - x')\partial'_j f(x') = \int_{V(\ell)} d^3x' \, \left[\partial'_j \Im_\ell(x - x')f(x') - f(x')\partial'_j \Im_\ell(x - x')\right] \, , \tag{7}$$

where we used the product rule. From the divergence theorem on the first term of Equation (7) follows:

$$\widetilde{\partial_j f(x)}^\ell = \int_{S(\ell)} d^2x' \, \hat{x}'_j \Im_\ell(x - x')f(x') - \int_{V(\ell)} d^3x' \, f(x')\partial'_j \Im_\ell(r) \, . \tag{8}$$

As the kernel is defined to be zero on the surface $S(\ell)$, the first term on the right-hand side of Equation (8) is zero. The derivative on the kernel thus gives:

$$\widetilde{\partial_j f(\boldsymbol{x})}^\ell = -\int_{V(\ell)} d^3x'\, f(\boldsymbol{x}')\ell^{-1}\frac{r_j}{|r_i|}\partial_U \Im_\ell(U)\,, \tag{9}$$

where we use $U = |r_j|/\ell$ and the derivative is $\partial_U \doteq \partial/\partial U$, introduced with the chain rule. We have also worked out that:

$$\partial'_j U \doteq \frac{\partial}{\partial x'_j}\frac{1}{\ell}(r_i r_i)^{1/2} = \frac{1}{2\ell}(r_i r_i)^{-1/2}\frac{\partial}{\partial x'_j}\left(x_i^2 + x_i'^2 - 2x'_i x_i\right)$$

$$= \frac{1}{2\ell}(r_i r_i)^{-1/2}2\delta_{ij}(x'_i - x_i) = \frac{1}{\ell}(r_i r_i)^{-1/2}r_j \doteq \frac{1}{\ell}\frac{r_j}{|r_i|} \tag{10}$$

$$\partial_j U \doteq \frac{\partial}{\partial x_j}\frac{1}{\ell}(r_i r_i)^{1/2} = \frac{1}{2\ell}(r_i r_i)^{-1/2}\frac{\partial}{\partial x_j}\left(x_i^2 + x_i'^2 - 2x'_i x_i\right)$$

$$= \frac{1}{2\ell}(r_i r_i)^{-1/2}2\delta_{ij}(x_i - x'_i) = \frac{-1}{\ell}(r_i r_i)^{-1/2}r_j \doteq \frac{-1}{\ell}\frac{r_j}{|r_i|}\,. \tag{11}$$

The definition of the Kronecker delta $\delta_{ij}$ was used, and the chain rule on the kernel is such: $\partial_j \Im_\ell(\boldsymbol{x} - \boldsymbol{x}') = -\partial'_j \Im_\ell(\boldsymbol{x} - \boldsymbol{x}')$. The derivative on the filtered field is thus:

$$\widetilde{\partial_j f(\boldsymbol{x})}^\ell = \partial_j \int_{V(\ell)} d^3x'\, \Im_\ell(\boldsymbol{x} - \boldsymbol{x}')f(\boldsymbol{x}') = \int_{V(\ell)} d^3x'\, f(\boldsymbol{x}')\partial_j \Im_\ell(\boldsymbol{x} - \boldsymbol{x}')\,. \tag{12}$$

This is a convolution integral, integrated over $d^3x'$, and by this definition:

$$\widetilde{\partial_j f(\boldsymbol{x})}^\ell = -\int_{V(\ell)} d^3x'\, f(\boldsymbol{x}')\ell^{-1}\frac{r_j}{|r_i|}\partial_U \Im_\ell(U)\,, \tag{13}$$

because $\partial_j f(\boldsymbol{x}') = 0$. The derivative of the kernel thus produces a minus sign this go-around. We have shown the filter and derivative commute under the assumption of integrability of the kernel and of the functions.

## 2.2. Cumulants

This allows us to generalize the cumulants of higher order, following from the definition of the kernel. The definition of the covariance (second order cumulant) is:

$$\tau_{ij}^\ell\big(\boldsymbol{f}(\boldsymbol{x}),\boldsymbol{g}(\boldsymbol{x})\big) = \int_{V(\ell)} d^3x'\, \Im_\ell(\boldsymbol{x} - \boldsymbol{x}')f_i(\boldsymbol{x}')\, g_j(\boldsymbol{x}')$$

$$- \left[\int_{V(\ell)} d^3x'\, \Im_\ell(\boldsymbol{x} - \boldsymbol{x}')f_i(\boldsymbol{x}')\right]\left[\int_{V(\ell)} d^3x'\, \Im_\ell(\boldsymbol{x} - \boldsymbol{x}')g_j(\boldsymbol{x}')\right], \tag{14}$$

where index of the tensor $\tau_{ij}^\ell$ follows to the parentheses in the order that they appear. It is possible to simplify the notation by defining:

$$\tau_{ij}^\ell\big(\boldsymbol{f}(\boldsymbol{x}),\boldsymbol{g}(\boldsymbol{x})\big) \doteq \tau_{ij}^\ell(\boldsymbol{f},\boldsymbol{g}) = \widetilde{f_i g_j} - \tilde{f}_i \tilde{g}_j\,. \tag{15}$$

The third-order cumulants appear:

$$\tau_{ijk}^\ell(\boldsymbol{f},\boldsymbol{g},\boldsymbol{h}) = \widetilde{f_i g_j h_k} - \tau_{ij}^\ell(\boldsymbol{f},\boldsymbol{g})\tilde{h}_k - \tau_{ik}^\ell(\boldsymbol{f},\boldsymbol{h})\tilde{g}_j - \tau_{jk}^\ell(\boldsymbol{g},\boldsymbol{h})\tilde{f}_i - \tilde{f}_i \tilde{g}_j \tilde{h}_k \tag{16}$$

$$= \widetilde{f_i g_j h_k} - \widetilde{f_i g_j}\,\tilde{h}_k - \widetilde{f_i h_k}\tilde{g}_j - \widetilde{g_j h_k}\tilde{f}_i + 2\tilde{f}_i \tilde{g}_j \tilde{h}_k\,. \tag{17}$$

This notation uses the order of the cumulant as the number of terms separated by commas and the rank of the tensor as the number of non-paired indexes.

## 2.3. Norms and Inequalities

Above, we define the Euclidean norm $|f_i| = \sqrt{f_i f_i}$. In the following, the Lebesque space norms of order $p$, which we call $L_p$-norms, are necessary. We define here such norm as:

$$||f_i||_p = \left[ \frac{1}{V} \int_V d^3x \, |f_i|^p \right]^{1/p} < \infty, \tag{18}$$

where the measure is the physical space $x$. Note that $||f_i||_\infty = \sup\{||f_i||_p\}$. We further introduce the Hölder inequality:

$$|| \Pi_j f_i^{(j)} ||_p \leq \Pi_j ||f_i^{(j)}||_{p_j}; \quad \sum_j \frac{1}{p_j} = p, \, p \geq 1, \tag{19}$$

where $\Pi_j$ is the product operation. Furthermore, we make use of the Minkowski inequality:

$$||f_i + g_i||_p \leq ||f_i||_p + ||g_i||_p, \tag{20}$$

and, finally, Young's inequality for convolutions, which in the notation of the present paper appears as:

$$||\widetilde{y}_i^\ell(x)||_r \leq ||\Im_\ell(x - x')||_p \, ||y_i(x')||_q; \quad 1 + \frac{1}{r} = \frac{1}{p} + \frac{1}{q}, \tag{21}$$

(note that proof of this equation does require integrating $d^3x$, $d^3x'$).

## 2.4. Estimates

We use the Hölder condition given above to estimate various terms in our derivation. In the Euclidean space, the condition can be stated as:

$$|f(x') - f(x)| \leq a \, |x' - x|^b, \tag{22}$$

for $a, b$ real positive numbers. The condition is understood, assuming $a < \infty$, so that for $0 < b < 1$ the function $f$ is continuous, for $b = 1$ the function $f$ is Lipschitz continuous and for $b > 1$ the function $f$ is constant. A sharp function appears discontinuous to an increment $|x' - x|$ that is too large, and as the increment is decreased it changes dramatically, once the sharpness is resolved, becoming again continuous. The sharper is the function, the more dramatic and smaller is the event that occurs. Sharper functions have smaller Hölder exponent $b$, for functions on the interval $0 < b < 1$.

## 3. Filtered Incompressible Magnetohydrodynamic Equation

The incompressible magnetohydrodynamic (IMHD) equations can be written as:

$$\partial_t y_i + \partial_j w_j y_i = -\partial_i p + \nu \partial_j^2 y_i \, ; \quad \partial_i y_i = 0 \, . \tag{23}$$

The vector field $y_i = u_i + b_i$, $w_i = u_i - b_i$ are the Elsasser variables, so that switching $y_i \to w_i$ and $w_i \to y_i$ recovers the two other equations, and sum/differences recover the momentum equations for the velocity $u_i$ and Alfvén velocity $b_i$, respectively. The dissipative coefficient $\nu$ requires resistivity equal to the viscosity (i.e., Prandtl number $P_r = 1$). Note that, as the limit $\nu \to 0$ is only considered in the

following, we are not interested in specific dissipation range physics, where the IMHD approximation fails. Writing the filtered IMHD equation:

$$\partial_t \tilde{y}_i + \partial_j \widetilde{w_j y_i} = -\partial_i \tilde{p} + \nu \partial_j^2 \tilde{y}_i \; ; \quad \partial_i \tilde{y}_i = 0 \, , \tag{24}$$

since the derivative and filter commute. We use the simplified notation:

$$\tilde{y}_i^\ell(x') \to \tilde{y}_i' \quad \tilde{y}_i^\ell(x) \to \tilde{y}_i \tag{25}$$

$$\bar{y}_i^\ell(x') \to \bar{y}_i' \quad \bar{y}_i^\ell(x) \to \bar{y}_i \tag{26}$$

$$\Im_\ell(x - x') \to \Im_\ell \tag{27}$$

## 4. Energy Budget

In these sections, we build energy budget equations for the large, small and total scales of the system. The technique essentially consists of writing an energy-like equation for the fields, writing all terms as fluxes through surfaces or internal change within the volume, and then sub-categorizing into scale-to-scale transfer when they appear in large- or small-scale equations, or inter-scale otherwise. We start by obtaining a tensor version of the equation, so that the role of anisotropy can be discussed. Additionally, we write the time-evolution equation for the cross-field, which is the non-linear term in the IMHD equation (Equation (23)).

### 4.1. Large-Scale Budget Equations

In these sections, we develop the large-scale budget equations. Let us define the large-scale densities:

$$\tilde{e}_{ij}^y = \frac{1}{2} \tilde{y}_i \tilde{y}_j, \; \tilde{e}_{ij}^w = \frac{1}{2} \tilde{w}_i \tilde{w}_j, \; \tilde{e}_{ij}^r = \tilde{y}_i \tilde{w}_j \, . \tag{28}$$

These quantities are the fields averaged over a volume characterized by arbitrary $\ell$, and represent the quantities associated to the large scale. Obviously, the true energies can be recovered by applying a Kronecker delta $\delta_{ij}$ and integrating:

$$\tilde{E}_{ii}^y = \frac{1}{V} \int_V d^3x \; \delta_{ij} \tilde{e}_{ij}^y \tag{29}$$

$$\tilde{E}_{ii}^w = \frac{1}{V} \int_V d^3x \; \delta_{ij} \tilde{e}_{ij}^w \tag{30}$$

$$\tilde{E}_{ii}^r = \frac{1}{V} \int_V d^3x \; \delta_{ij} \tilde{e}_{ij}^r \tag{31}$$

#### 4.1.1. Energy

Here, we develop an equation for $\partial_t \tilde{y}_i \tilde{y}_j$. As done before, we write the IMHD equations:

$$\partial_t \tilde{y}_i + \partial_k \widetilde{w_k y_i} = -\partial_i \tilde{p} + \nu \partial_k^2 \tilde{y}_i \, , \tag{32}$$

introduce the second order cumulant for the non-linear terms:

$$\partial_t \tilde{y}_i + \partial_k \left( \tau_{ik}^\ell(y, w) + \tilde{w}_k \tilde{y}_i \right) = -\partial_i \tilde{p} + \nu \partial_k^2 \tilde{y}_i \, , \tag{33}$$

then take the inner product with $\tilde{y}_j$:

$$\tilde{y}_j \partial_t \tilde{y}_i + \tilde{y}_j \partial_k \left( \tau_{ik}^\ell(y, w) + \tilde{w}_k \tilde{y}_i \right) = -\tilde{y}_j \partial_i \tilde{p} + \nu \tilde{y}_j \partial_k^2 \tilde{y}_i \, , \tag{34}$$

and perform the same calculation for the $\widetilde{y}_i \partial_t \widetilde{y}_j = \dots$ to eventually produce:

$$\widetilde{y}_i \partial_t \widetilde{y}_j + \widetilde{y}_i \partial_k \left( \tau_{jk}^{\ell}(\boldsymbol{y}, \boldsymbol{w}) + \widetilde{w}_k \widetilde{y}_j \right) = -\widetilde{y}_i \partial_j \widetilde{p} + \nu \widetilde{y}_i \partial_k^2 \widetilde{y}_j \,. \tag{35}$$

Summing the two above equations, and using incompressibility and product rule to recombine the terms, we obtain:

$$\partial_t \widetilde{y}_i \widetilde{y}_j + \partial_k \widetilde{y}_j \tau_{ik}^{\ell}(\boldsymbol{y}, \boldsymbol{w}) - \tau_{ik}^{\ell}(\boldsymbol{y}, \boldsymbol{w}) \partial_k \widetilde{y}_j + \partial_k \widetilde{y}_i \tau_{jk}^{\ell}(\boldsymbol{y}, \boldsymbol{w}) - \tau_{jk}^{\ell}(\boldsymbol{y}, \boldsymbol{w}) \partial_k \widetilde{y}_i + \partial_k \widetilde{w}_k \widetilde{y}_i \widetilde{y}_j$$
$$= -\left( \widetilde{y}_j \partial_i + \widetilde{y}_i \partial_j \right) \widetilde{p} + \nu \left[ \partial_k^2 \widetilde{y}_i \widetilde{y}_j - 2 (\partial_k \widetilde{y}_i)(\partial_k \widetilde{y}_j) \right] \,. \tag{36}$$

Note that we unravel d the term containing the second-order cumulant, as it is convenient below. The same procedure for the $\widetilde{w}$-field produces:

$$\partial_t \widetilde{w}_i \widetilde{w}_j + \partial_k \widetilde{w}_j \tau_{ik}^{\ell}(\boldsymbol{w}, \boldsymbol{y}) - \tau_{ik}^{\ell}(\boldsymbol{w}, \boldsymbol{y}) \partial_k \widetilde{w}_j + \partial_k \widetilde{w}_i \tau_{jk}^{\ell}(\boldsymbol{w}, \boldsymbol{y}) - \tau_{jk}^{\ell}(\boldsymbol{w}, \boldsymbol{y}) \partial_k \widetilde{w}_i + \partial_k \widetilde{y}_k \widetilde{w}_i \widetilde{w}_j$$
$$= -\left( \widetilde{w}_j \partial_i + \widetilde{w}_i \partial_j \right) \widetilde{p} + \nu \left[ \partial_k^2 \widetilde{w}_i \widetilde{w}_j - 2 (\partial_k \widetilde{w}_i)(\partial_k \widetilde{w}_j) \right] \,. \tag{37}$$

### 4.1.2. Cross-Field

We are now interested in the evolution of a second order term $\widetilde{r}_{ij} = \widetilde{y}_i \widetilde{w}_j$, often referred to as the residual energy density for $j = i$. Thus, we write an equation for each, take the scalar product with the other field, for the two equations:

$$\widetilde{w}_j \partial_t \widetilde{y}_i + \widetilde{w}_j \partial_k \left( \tau_{ik}^{\ell}(\boldsymbol{y}, \boldsymbol{w}) + \widetilde{y}_i \widetilde{w}_k \right) = -\widetilde{w}_j \partial_i \widetilde{p} + \nu \widetilde{w}_j \partial_k^2 \widetilde{y}_i \tag{38}$$

$$\widetilde{y}_i \partial_t \widetilde{w}_j + \widetilde{y}_i \partial_k \left( \tau_{jk}^{\ell}(\boldsymbol{w}, \boldsymbol{y}) + \widetilde{w}_j \widetilde{y}_k \right) = -\widetilde{y}_i \partial_j \widetilde{p} + \nu \widetilde{y}_i \partial_k^2 \widetilde{w}_j \,. \tag{39}$$

After summing the equations and some simplifications:

$$\partial_t \widetilde{y}_i \widetilde{w}_j + \widetilde{w}_j \partial_k \left( \tau_{ik}^{\ell}(\boldsymbol{y}, \boldsymbol{w}) + \widetilde{y}_i \widetilde{w}_k \right) + \widetilde{y}_i \partial_k \left( \tau_{jk}^{\ell}(\boldsymbol{w}, \boldsymbol{y}) + \widetilde{w}_j \widetilde{y}_k \right) =$$
$$-\widetilde{w}_j \partial_i \widetilde{p} - \widetilde{y}_i \partial_j \widetilde{p} + \nu \left[ \widetilde{w}_j \partial_k^2 \widetilde{y}_i + \widetilde{y}_i \partial_k^2 \widetilde{w}_j \right] \,. \tag{40}$$

This is the temporal evolution of the non-linear terms in the IMHD equations. Notice that here swapping $i, j$ does change the equation.

### 4.2. Small-Scale Energy Budget

We now seek the-small scale energy equations; most intuitively, we define the total energy in the $y_i$-field: $\frac{1}{2} \int_V d^3x \, y_i^2$ so that the small scale energy is:

$$\frac{1}{2} \int_V d^3x \, \bar{y}_i^2 = \frac{1}{2} \int_V d^3x \left[ y_i^2 - \widetilde{y}_i^2 \right] \,, \tag{41}$$

as obtained by subtracting off the large-scale contribution. With the definition of the filtered equation $\int_V d^3x \widetilde{f} = \int_V d^3x f$:

$$\frac{1}{2} \int_V d^3x \, \bar{y}_i^2 = \frac{1}{2} \int_V d^3x \left[ \widetilde{\widehat{y_i^2}} - \widetilde{y}_i^2 \right] \,. \tag{42}$$

This is the second-order cumulant $\tau_{ii}^{\ell}(\boldsymbol{y}, \boldsymbol{y})$, as defined above. We introduce now some definitions relative to the small-scale energy density:

$$\bar{e}_{ij}^y = \frac{1}{2} \tau_{ij}^{\ell}(\boldsymbol{y}, \boldsymbol{y}), \ \bar{e}_{ij}^w = \frac{1}{2} \tau_{ij}^{\ell}(\boldsymbol{w}, \boldsymbol{w}), \ \bar{e}_{ij}^r = \tau_{ij}^{\ell}(\boldsymbol{y}, \boldsymbol{w}) \,, \tag{43}$$

and the actual small-scale energies:

$$\bar{E}_{ii}^y = \frac{1}{V} \int_V d^3x \, \delta_{ij} \bar{e}_{ij}^y \tag{44}$$

$$\bar{E}_{ii}^w = \frac{1}{V} \int_V d^3x \, \delta_{ij} \bar{e}_{ij}^w \tag{45}$$

$$\bar{E}_{ii}^r = \frac{1}{V} \int_V d^3x \, \delta_{ij} \bar{e}_{ij}^r . \tag{46}$$

Thus, we seek a time-evolution equation for the cumulant: $\tau_{ij}^\ell(\boldsymbol{y}, \boldsymbol{y})$, $\tau_{ij}^\ell(\boldsymbol{w}, \boldsymbol{w})$ $\tau_{ij}^\ell(\boldsymbol{y}, \boldsymbol{w})$.

### 4.2.1. Small-Scale Cross-Field

Let us start by writing an equation for $\partial_t y_i w_j$ using the IMHD Equation (23) as done above, but without using the filter. Taking the inner product with each other to couple them, and then summing produces:

$$\partial_t y_i w_j + w_j \partial_k w_k y_i + y_i \partial_k y_k w_j = -y_i \partial_j p - w_j \partial_i p + \nu y_i \partial_k^2 w_j + \nu w_j \partial_k^2 y_i . \tag{47}$$

This can be rewritten as:

$$\partial_t y_i w_j + \partial_k w_j w_k y_i + \partial_k y_i y_k w_j - y_i \partial_k w_k w_j - w_j \partial_k y_k y_i = -y_i \partial_j p - w_j \partial_i p + \nu y_i \partial_k^2 w_j + \nu w_j \partial_k^2 y_i . \tag{48}$$

By using the product rule on the pressure and viscous terms:

$$\begin{aligned} \partial_t y_i w_j + \partial_k w_j w_k y_i + \partial_k y_i y_k w_j - y_i \partial_k w_k w_j - w_j \partial_k y_k y_i \\ = -\partial_j y_i p + p \partial_j y_i - \partial_i w_j p + p \partial_i w_j - 2\nu (\partial_k y_i)(\partial_k w_j) + \nu \partial_k^2 w_j y_i . \end{aligned} \tag{49}$$

Now, let us apply the filter:

$$\begin{aligned} \partial_t \widetilde{y_i w_j} + \partial_k \widetilde{w_j w_k y_i} + \partial_k \widetilde{y_i y_k w_j} - \widetilde{y_i \partial_k w_k w_j} - \widetilde{w_j \partial_k y_k y_i} \\ = -\partial_j \widetilde{y_i p} + \widetilde{p \partial_j y_i} - \partial_i \widetilde{w_j p} + \widetilde{p \partial_i w_j} - 2\nu \widetilde{(\partial_k y_i)(\partial_k w_j)} + \nu \partial_k^2 \widetilde{w_j y_i} . \end{aligned} \tag{50}$$

Next, we subtract the time evolution equation for $\partial_t \tilde{y}_i \tilde{w}_j$ (Equation (40)), but after shuffling the third-order terms with product rule as:

$$\begin{aligned} \partial_t \tilde{y}_i \tilde{w}_j + \partial_k \tilde{y}_i \tau_{jk}^\ell(\boldsymbol{w}, \boldsymbol{y}) - \tau_{jk}^\ell(\boldsymbol{w}, \boldsymbol{y}) \partial_k \tilde{y}_i + \partial_k \tilde{y}_i \tilde{y}_k \tilde{w}_j - \tilde{w}_j \partial_k \tilde{y}_k \tilde{y}_i \\ + \partial_k \tilde{w}_j \tau_{ik}^\ell(\boldsymbol{y}, \boldsymbol{w}) - \tau_{ik}^\ell(\boldsymbol{y}, \boldsymbol{w}) \partial_k \tilde{w}_j + \partial_k \tilde{w}_j \tilde{w}_k \tilde{y}_i - \tilde{y}_i \partial_k \tilde{w}_k \tilde{w}_j \\ = -\partial_i \tilde{w}_j \tilde{p} + \tilde{p} \partial_i \tilde{w}_j - \partial_j \tilde{y}_i \tilde{p} + \tilde{p} \partial_j \tilde{y}_i + \nu \partial_k^2 \tilde{y}_i \tilde{w}_j - 2\nu (\partial_k \tilde{y}_i)(\partial_k \tilde{w}_j) . \end{aligned} \tag{51}$$

We can now write the second order cumulants and notice cancellation of therms:

$$\begin{aligned} \partial_t \tilde{y}_i \tilde{w}_j + \partial_k \tilde{y}_i \widetilde{w_j y_k} - \widetilde{w_j y_k} \partial_k \tilde{y}_i + \partial_k \tilde{w}_j \widetilde{y_i w_k} - \widetilde{y_i w_k} \partial_k \tilde{w}_j \\ = -\partial_i \tilde{w}_j \tilde{p} + \tilde{p} \partial_i \tilde{w}_j - \partial_j \tilde{y}_i \tilde{p} + \tilde{p} \partial_j \tilde{y}_i + \nu \partial_k^2 \tilde{y}_i \tilde{w}_j - 2\nu (\partial_k \tilde{y}_i)(\partial_k \tilde{w}_j) . \end{aligned} \tag{52}$$

Now, we want to write an equation for $\partial_t \tau_{ij}^\ell(\boldsymbol{y}, \boldsymbol{w})$, thus we write Equation (50) minus Equation (52) as:

$$\partial_t \tau_{ij}^\ell(\mathbf{y}, \mathbf{w}) + \partial_k \widetilde{\overline{w_j w_k y_i}} + \partial_k \widetilde{\overline{y_i y_k w_j}} - \widetilde{\overline{y_i \partial_k w_k w_j}} - \widetilde{\overline{w_j \partial_k y_k y_i}}$$
$$- \partial_k \widetilde{y_i} \widetilde{\overline{w_j y_k}} + \widetilde{\overline{w_j y_k}} \partial_k \widetilde{y_i} - \partial_k \widetilde{w_j} \widetilde{\overline{y_i w_k}} + \widetilde{\overline{y_i w_k}} \partial_k \widetilde{w_j}$$
$$= -\partial_j \tau_i^\ell(\mathbf{y}, p) + \tau_{ji}^\ell(p, \nabla \mathbf{y}) - \partial_i \tau_j^\ell(\mathbf{w}, p) + \tau_{ij}^\ell(p, \nabla \mathbf{w})$$
$$+ \nu \partial_k^2 \tau_{ij}^\ell(\mathbf{y}, \mathbf{w}) - 2\nu \tau_{kikj}^\ell(\nabla \mathbf{y}, \nabla \mathbf{w}) \,. \tag{53}$$

Notice that after using incompressibility on the fourth in the first line of last equation, it can be combined with he fourth term line two, for a second-order cumulant, and then the same can be applied to the fifth term in line one with the second term on line two, so that:

$$\partial_t \tau_{ij}^\ell(\mathbf{y}, \mathbf{w}) + \partial_k \widetilde{\overline{w_j w_k y_i}} + \partial_k \widetilde{\overline{y_i y_k w_j}} - \tau_{ikkj}^\ell(\mathbf{yw}, \nabla \mathbf{w}) - \tau_{jkki}^\ell(\mathbf{wy}, \nabla \mathbf{y}) - \partial_k \widetilde{y_i} \widetilde{\overline{w_j y_k}} - \partial_k \widetilde{w_j} \widetilde{\overline{y_i w_k}}$$
$$= -\partial_j \tau_i^\ell(\mathbf{y}, p) + \tau_{ji}^\ell(p, \nabla \mathbf{y}) - \partial_i \tau_j^\ell(\mathbf{w}, p) + \tau_{ij}^\ell(p, \nabla \mathbf{w}) + \nu \partial_k^2 \tau_{ij}^\ell(\mathbf{y}, \mathbf{w}) - 2\nu \tau_{kikj}^\ell(\nabla \mathbf{y}, \nabla \mathbf{w}) \,. \tag{54}$$

We can now introduce two more second-order cumulants to combine the remaining terms:

$$\partial_t \tau_{ij}^\ell(\mathbf{y}, \mathbf{w}) + \partial_k \tau_{ikj}^\ell(\mathbf{yw}, \mathbf{w}) + \partial_k \tau_{kji}^\ell(\mathbf{yw}, \mathbf{y}) - \tau_{ikkj}^\ell(\mathbf{yw}, \nabla \mathbf{w}) - \tau_{jkki}^\ell(\mathbf{wy}, \nabla \mathbf{y})$$
$$= -\partial_j \tau_i^\ell(\mathbf{y}, p) + \tau_{ji}^\ell(p, \nabla \mathbf{y}) - \partial_i \tau_j^\ell(\mathbf{w}, p) + \tau_{ij}^\ell(p, \nabla \mathbf{w}) + \nu \partial_k^2 \tau_{ij}^\ell(\mathbf{y}, \mathbf{w}) - 2\nu \tau_{kikj}^\ell(\nabla \mathbf{y}, \nabla \mathbf{w}) \,. \tag{55}$$

Note that setting the fields as equal, $y_i = w_i = u_i$, gives the hydrodynamic version of the equation, and it is easy to show this reduces to the published result in Germano [5]. Finally, we write here another form of this result, using results of the cumulants, which are needed below:

$$\partial_t \tau_{ij}^\ell(\mathbf{y}, \mathbf{w}) + \partial_k \tau_{ikj}^\ell(\mathbf{yw}, \mathbf{w}) + \partial_k \tau_{kji}^\ell(\mathbf{yw}, \mathbf{y}) - \widetilde{\overline{y_i w_k \partial_k w_j}} + \tau_{ik}^\ell(\mathbf{y}, \mathbf{w}) \partial_k \widetilde{w}_j$$
$$+ \widetilde{y}_i \widetilde{w}_k \partial_k \widetilde{w}_j - \widetilde{\overline{w_j y_k \partial_k y_j}} + \tau_{jk}^\ell(\mathbf{w}, \mathbf{y}) \partial_k \widetilde{y}_i + \widetilde{w}_j \widetilde{y}_k \partial_k \widetilde{y}_i = -\partial_j \tau_i^\ell(\mathbf{y}, p)$$
$$+ \tau_{ji}^\ell(p, \nabla \mathbf{y}) - \partial_i \tau_j^\ell(\mathbf{w}, p) + \tau_{ij}^\ell(p, \nabla \mathbf{w}) + \nu \partial_k^2 \tau_{ij}^\ell(\mathbf{y}, \mathbf{w}) - 2\nu \tau_{kikj}^\ell(\nabla \mathbf{y}, \nabla \mathbf{w}) \,. \tag{56}$$

### 4.2.2. Small-Scale Energy

We write the small-scale equations for $\tau_{ij}^\ell(\mathbf{y}, \mathbf{y})$, $\tau_{ij}^\ell(\mathbf{w}, \mathbf{w})$ in the same manner as done above. Starting with the $y$-field, we look for an evolution equation for $\partial_t \widetilde{y_i y_j}$. Writing the IMHD equation,

$$\partial_t y_i + \partial_k w_k y_i = -\partial_i p + \nu \partial_j^2 y_i \,, \tag{57}$$

taking the inner product with $y_j$:

$$y_j \partial_t y_i + y_j \partial_k w_k y_i = -y_j \partial_i p + \nu y_j \partial_k^2 y_i \,, \tag{58}$$

and making the reverse calculation leads to:

$$y_i \partial_t y_j + y_i \partial_k w_k y_j = -y_i \partial_j p + \nu y_i \partial_k^2 y_j \,. \tag{59}$$

Summing those equations and using incompressibility and product rule gives:

$$\partial_t y_i y_j + \partial_k w_k y_i y_j = -(y_j \partial_i + y_i \partial_j) p + \nu \left[ \partial_k^2 y_i y_j - 2(\partial_k y_i)(\partial_k y_j) \right] \,. \tag{60}$$

Applying the filter gives:

$$\partial_t \widetilde{y_i y_j} + \partial_k \widetilde{w_k y_i y_j} = -\widetilde{y_j \partial_i p} - \widetilde{y_i \partial_j p} + \nu \left[ \partial_k^2 \widetilde{y_i y_j} - 2\widetilde{(\partial_k y_i)(\partial_k y_j)} \right] \,. \tag{61}$$

Let us re-write the large-scale equation and rework some of the terms with incompressibility and product rule:

$$\partial_t \widetilde{y}_i \widetilde{y}_j + \partial_k \widetilde{y}_j \tau_{ik}^\ell(\boldsymbol{y}, \boldsymbol{w}) - \tau_{ik}^\ell(\boldsymbol{y}, \boldsymbol{w}) \partial_k \widetilde{y}_j + \partial_k \widetilde{y}_i \tau_{jk}^\ell(\boldsymbol{y}, \boldsymbol{w}) - \tau_{jk}^\ell(\boldsymbol{y}, \boldsymbol{w}) \partial_k \widetilde{y}_i + \partial_k \widetilde{w}_k \widetilde{y}_i \widetilde{y}_j$$

$$= -\left(\widetilde{y}_j \partial_i + \widetilde{y}_i \partial_j\right) \widetilde{p} + \nu \left[\partial_k^2 \widetilde{y}_i \widetilde{y}_j - 2(\partial_k \widetilde{y}_i)(\partial_k \widetilde{y}_j)\right]. \tag{62}$$

Subtracting this equation from Equation (61), we can write the desired time evolution equation:

$$\partial_t \tau_{ij}^\ell(\boldsymbol{y}, \boldsymbol{y}) - \partial_k \widetilde{y}_j \tau_{ik}^\ell(\boldsymbol{y}, \boldsymbol{w}) + \tau_{ik}^\ell(\boldsymbol{y}, \boldsymbol{w}) \partial_k \widetilde{y}_j - \partial_k \widetilde{y}_i \tau_{jk}^\ell(\boldsymbol{y}, \boldsymbol{w}) + \tau_{jk}^\ell(\boldsymbol{y}, \boldsymbol{w}) \partial_k \widetilde{y}_i - \partial_k \widetilde{w}_k \widetilde{y}_i \widetilde{y}_j + \partial_k \widetilde{w_k y_i y_j}$$

$$= -\tau_{ji}^\ell(\boldsymbol{y}, \nabla p) - \tau_{ij}^\ell(\boldsymbol{y}, \nabla p) + \nu \left[\partial_k^2 \tau_{ij}^\ell(\boldsymbol{y}, \boldsymbol{y}) - 2\tau_{kikj}^\ell(\nabla \boldsymbol{y}, \nabla \boldsymbol{y})\right]. \tag{63}$$

Notice that terms two, four, six and seven can be rewritten as a third-order cumulant and an additional term:

$$\partial_t \tau_{ij}^\ell(\boldsymbol{y}, \boldsymbol{y}) + \partial_k \tau_{ijk}^\ell(\boldsymbol{y}, \boldsymbol{y}, \boldsymbol{w}) + \partial_k \widetilde{w}_k \tau_{ij}(\boldsymbol{y}, \boldsymbol{y}) + \tau_{ik}^\ell(\boldsymbol{y}, \boldsymbol{w}) \partial_k \widetilde{y}_j + \tau_{jk}^\ell(\boldsymbol{y}, \boldsymbol{w}) \partial_k \widetilde{y}_i$$

$$= -\tau_{ji}^\ell(\boldsymbol{y}, \nabla p) - \tau_{ij}^\ell(\boldsymbol{y}, \nabla p) + \nu \left[\partial_k^2 \tau_{ij}^\ell(\boldsymbol{y}, \boldsymbol{y}) - 2\tau_{kikj}^\ell(\nabla \boldsymbol{y}, \nabla \boldsymbol{y})\right]. \tag{64}$$

Finally, repeating the same procedure for the *w*-field, which is equivalent to substituting $y_i \rightarrow w_i,\ y_j \rightarrow w_j,\ w_k \rightarrow y_k$, we obtain the following equation:

$$\partial_t \tau_{ij}^\ell(\boldsymbol{w}, \boldsymbol{w}) + \partial_k \tau_{ijk}^\ell(\boldsymbol{w}, \boldsymbol{w}, \boldsymbol{y}) + \partial_k \widetilde{y}_k \tau_{ij}(\boldsymbol{w}, \boldsymbol{w}) + \tau_{ik}^\ell(\boldsymbol{w}, \boldsymbol{y}) \partial_k \widetilde{w}_j + \tau_{jk}^\ell(\boldsymbol{w}, \boldsymbol{y}) \partial_k \widetilde{w}_i$$

$$= -\tau_{ji}^\ell(\boldsymbol{w}, \nabla p) - \tau_{ij}^\ell(\boldsymbol{w}, \nabla p) + \nu \left[\partial_k^2 \tau_{ij}^\ell(\boldsymbol{w}, \boldsymbol{w}) - 2\tau_{kikj}^\ell(\nabla \boldsymbol{w}, \nabla \boldsymbol{w})\right]. \tag{65}$$

### 4.3. Ideal Turbulence

Once the large/small -scale equations have been written, it is necessary to introduce a way to distinguish between scale-to-scale, inter-scale and dissipative energy transfers. The introduction of the filter ensures the following convenient property of the viscous terms:

$$\lim_{\nu \to 0} \nu \partial_j^2 \widetilde{w}_i = 0. \tag{66}$$

To demonstrate the above result, we make use of the commutation of operations, the properties of the convolution and the previously described result on the derivative of the kernel:

$$\partial_j^2 \widetilde{w}_i = \int_{V(\ell)} d^3 x'\ w_i' \partial_j^2 \Im_\ell = \int_{V(\ell)} d^3 x'\ w_i' \ell^{-2} \partial_U^2 \Im_\ell(U). \tag{67}$$

As square-integrable real valued functions are concerned here, the Cauchy–Schwartz inequality provides:

$$\left| \int_{V(\ell)} d^3 x'\ w_i' \ell^{-2} \partial_U^2 \Im_\ell(U) \right|^2 \leq \int_{V(\ell)} d^3 x'\ |w_i'|^2 \int_{V(\ell)} d^3 x'\ \left| \ell^{-2} \partial_U^2 \Im_\ell(U) \right|^2. \tag{68}$$

The integral that contains the kernel is constant if the latter is differentiable and square-integrable. Let us set the entire integral term to a constant:

$$\int_{V(\ell)} d^3 x'\ \left| \ell^{-2} \partial_U^2 \Im_\ell(U) \right|^2 \rightarrow C_\ell \implies \left| \int_{V(\ell)} d^3 x'\ w_i' \ell^{-2} \partial_U^2 \Im_\ell(U) \right|^2 \leq C_\ell \int_{V(\ell)} d^3 x'\ |w_i'|^2. \tag{69}$$

If the integral on the right-hand side of the above inequality is finite, the left-hand side must be finite. This proves that:

$$\lim_{\nu \to 0} \nu \partial_j^2 \widetilde{w}_i = 0 \,. \tag{70}$$

This result affirms that, for *ideal turbulence* ($\nu \to 0$), if the generalized energy density $\sim |w_i|^2$ stays finite, then the details of viscous dissipation can be disregarded without singularities in the large-scale equations. Similarly, when writing the IMHD equation and applying the filter:

$$\lim_{\nu \to 0} \partial_k^2 \widetilde{y}_i \widetilde{w}_i = 0 \,, \tag{71}$$

then write the Elsasser variables explicitly in terms of the primitive variables:

$$\partial_k^2 \widetilde{y}_i \widetilde{w}_i = \partial_k^2 \widetilde{v}_i^2 - \partial_k^2 \widetilde{b}_i^2 \,, \tag{72}$$

so that if the energy in both the velocity and magnetic fields stays finite, the limit does not create singularities. This result can be generalized to the anisotropic case by demanding that the integral over the individual fields themselves stay finite. The only term that can not be disregarded appears in the small-scale terms: $\tau_{kikj}^\ell(\nabla f, \nabla g)$ for both $f = g$, $f \neq g$.

### 4.4. Ideal IMHD Equations

We now proceed with the IMHD version. Let us write the ideal, large-scale equations in the ideal case, namely after taking the limit $\nu \to 0$ in the large-scale equations:

$$
\begin{aligned}
\partial_t \widetilde{y}_i \widetilde{y}_j &+ \partial_k \widetilde{y}_j \tau_{ik}^\ell(\boldsymbol{y}, \boldsymbol{w}) + \partial_k \widetilde{y}_i \tau_{jk}^\ell(\boldsymbol{y}, \boldsymbol{w}) - \partial_k \widetilde{w}_k \widetilde{y}_i \widetilde{y}_j \\
&= \tau_{jk}^\ell(\boldsymbol{y}, \boldsymbol{w}) \partial_k \widetilde{y}_i + \tau_{ik}^\ell(\boldsymbol{y}, \boldsymbol{w}) \partial_k \widetilde{y}_j - \left( \widetilde{y}_j \partial_i + \widetilde{y}_i \partial_j \right) \widetilde{p}
\end{aligned}
\tag{73}
$$

$$
\begin{aligned}
\partial_t \widetilde{w}_i \widetilde{w}_j &+ \partial_k \widetilde{w}_j \tau_{ik}^\ell(\boldsymbol{w}, \boldsymbol{y}) + \partial_k \widetilde{w}_i \tau_{jk}^\ell(\boldsymbol{w}, \boldsymbol{y}) - \partial_k \widetilde{y}_k \widetilde{w}_i \widetilde{w}_j \\
&= \tau_{jk}^\ell(\boldsymbol{w}, \boldsymbol{y}) \partial_k \widetilde{w}_i + \tau_{ik}^\ell(\boldsymbol{w}, \boldsymbol{y}) \partial_k \widetilde{w}_j - \left( \widetilde{w}_j \partial_i + \widetilde{w}_i \partial_j \right) \widetilde{p}
\end{aligned}
\tag{74}
$$

$$
\begin{aligned}
\partial_t \widetilde{y}_i \widetilde{w}_j &+ \partial_k \widetilde{y}_i \tau_{jk}^\ell(\boldsymbol{w}, \boldsymbol{y}) + \partial_k \widetilde{y}_i \widetilde{y}_k \widetilde{w}_j + \partial_k \widetilde{w}_j \tau_{ik}^\ell(\boldsymbol{y}, \boldsymbol{w}) + \partial_k \widetilde{w}_j \widetilde{w}_k \widetilde{y}_i + \partial_i \widetilde{w}_j \widetilde{p} + \partial_j \widetilde{y}_i \widetilde{p} \\
&= \widetilde{y}_i \partial_k \widetilde{w}_k \widetilde{w}_j + \tau_{ik}^\ell(\boldsymbol{y}, \boldsymbol{w}) \partial_k \widetilde{w}_j + \tau_{jk}^\ell(\boldsymbol{w}, \boldsymbol{y}) \partial_k \widetilde{y}_i + \widetilde{w}_j \partial_k \widetilde{y}_k \widetilde{y}_i + \widetilde{p} \partial_i \widetilde{w}_j + \widetilde{p} \partial_j \widetilde{y}_i \,.
\end{aligned}
\tag{75}
$$

The previous equations include some rearranging for the sake of clarity, and writing in one of the several forms that can be written. By multiplying the first two equations by $1/2$ and integrating each equation in space $V^{-1} \int_V d^3x$ to disregard surface flux terms, one obtains:

$$\partial_t \frac{1}{V} \int_V d^3x \, \widetilde{e}_{ij}^y = \frac{1}{2V} \int_V d^3x \left[ \tau_{jk}^\ell(\boldsymbol{y}, \boldsymbol{w}) \partial_k \widetilde{y}_i + \tau_{ik}^\ell(\boldsymbol{y}, \boldsymbol{w}) \partial_k \widetilde{y}_j - \left( \widetilde{y}_j \partial_i + \widetilde{y}_i \partial_j \right) \widetilde{p} \right] \tag{76}$$

$$\partial_t \frac{1}{V} \int_V d^3x \, \widetilde{e}_{ij}^w = \frac{1}{2V} \int_V d^3x \left[ \tau_{jk}^\ell(\boldsymbol{w}, \boldsymbol{y}) \partial_k \widetilde{w}_i + \tau_{ik}^\ell(\boldsymbol{w}, \boldsymbol{y}) \partial_k \widetilde{w}_j - \left( \widetilde{w}_j \partial_i + \widetilde{w}_i \partial_j \right) \widetilde{p} \right] \tag{77}$$

$$
\begin{aligned}
\partial_t \frac{1}{V} \int_V d^3x \, \widetilde{e}_{ij}^r = \frac{1}{V} \int_V d^3x \Big[ &\widetilde{y}_i \partial_k \widetilde{w}_k \widetilde{w}_j + \tau_{ik}^\ell(\boldsymbol{y}, \boldsymbol{w}) \partial_k \widetilde{w}_j + \tau_{jk}^\ell(\boldsymbol{w}, \boldsymbol{y}) \partial_k \widetilde{y}_i \\
&+ \widetilde{w}_j \partial_k \widetilde{y}_k \widetilde{y}_i + \widetilde{p} \partial_i \widetilde{w}_j + \widetilde{p} \partial_j \widetilde{y}_i \Big] \,.
\end{aligned}
\tag{78}
$$

Note that here we use the above definitions in Equation (28) to redefine the equations. The same procedure on the small-scale energies provides:

$$\partial_t \frac{1}{V}\int_V d^3x\, \bar{e}^y_{ij} = \frac{1}{2V}\int_V d^3x \left[ -\tau^\ell_{ik}(\boldsymbol{y},\boldsymbol{w})\partial_k\widetilde{y}_j - \tau^\ell_{jk}(\boldsymbol{y},\boldsymbol{w})\partial_k\widetilde{y}_i \right.$$
$$\left. -\tau^\ell_{ji}(\boldsymbol{y},\nabla p) - \tau^\ell_{ij}(\boldsymbol{y},\nabla p) - 2\nu\tau^\ell_{kikj}(\nabla\boldsymbol{y},\nabla\boldsymbol{y}) \right] \tag{79}$$

$$\partial_t \frac{1}{V}\int_V d^3x\, \bar{e}^w_{ij} = \frac{1}{2V}\int_V d^3x \left[ -\tau^\ell_{ik}(\boldsymbol{w},\boldsymbol{y})\partial_k\widetilde{w}_j - \tau^\ell_{jk}(\boldsymbol{w},\boldsymbol{y})\partial_k\widetilde{w}_i \right.$$
$$\left. -\tau^\ell_{ji}(\boldsymbol{w},\nabla p) - \tau^\ell_{ij}(\boldsymbol{w},\nabla p) - 2\nu\tau^\ell_{kikj}(\nabla\boldsymbol{w},\nabla\boldsymbol{w}) \right] \tag{80}$$

$$\partial_t \frac{1}{V}\int_V d^3x\, \bar{r}^e_{ij} = \frac{1}{V}\int_V d^3x \left[ \widetilde{y_i w_k \partial_k w_j} - \tau^\ell_{ik}(\boldsymbol{y},\boldsymbol{w})\partial_k\widetilde{w}_j - \widetilde{y}_i\partial_k\widetilde{w}_k\widetilde{w}_j + \widetilde{w_j y_k \partial_k y_i} \right.$$
$$\left. -\tau^\ell_{jk}(\boldsymbol{w},\boldsymbol{y})\partial_k\widetilde{y}_i - \widetilde{w}_j\partial_k\widetilde{y}_k\widetilde{y}_i + \tau^\ell_{ji}(p,\nabla\boldsymbol{y}) + \tau^\ell_{ij}(p,\nabla\boldsymbol{w}) - 2\nu\tau^\ell_{kikj}(\nabla\boldsymbol{y},\nabla\boldsymbol{w}) \right]. \tag{81}$$

The scale-to-scale transfers can now be identified through the terms appearing in the large-scale and small scale equations, but with opposite sign. We define them so they correspond to transfer from large to small scale:

$$\Pi^y_{ij} = \frac{1}{2}\left[ -\tau^\ell_{ik}(\boldsymbol{y},\boldsymbol{w})\partial_k\widetilde{y}_j - \tau^\ell_{jk}(\boldsymbol{y},\boldsymbol{w})\partial_k\widetilde{y}_i + (\widetilde{y}_j\partial_i + \widetilde{y}_i\partial_j)\widetilde{p} \right] \tag{82}$$

$$\Pi^w_{ij} = \frac{1}{2}\left[ -\tau^\ell_{ik}(\boldsymbol{w},\boldsymbol{y})\partial_k\widetilde{w}_j - \tau^\ell_{jk}(\boldsymbol{w},\boldsymbol{y})\partial_k\widetilde{w}_i + (\widetilde{w}_j\partial_i + \widetilde{w}_i\partial_j) \right] \tag{83}$$

$$\Pi^r_{ij} = -\tau^\ell_{ik}(\boldsymbol{y},\boldsymbol{w})\partial_k\widetilde{w}_j - \widetilde{y}_i\partial_k\widetilde{w}_k\widetilde{w}_j - \tau^\ell_{jk}(\boldsymbol{w},\boldsymbol{y})\partial_k\widetilde{y}_i$$
$$- \widetilde{w}_j\partial_k\widetilde{y}_k\widetilde{y}_i - \widetilde{p}\partial_i\widetilde{w}_j - \widetilde{p}\partial_j\widetilde{y}_i. \tag{84}$$

The remaining non-viscous terms in each equation correspond to inter-scale transfers:

$$\Xi^y_{ij} = -\frac{1}{2}\left[ \widetilde{y_i\partial_j p} + \widetilde{y_j\partial_i p} \right] \tag{85}$$

$$\Xi^w_{ij} = -\frac{1}{2}\left[ \widetilde{w_i\partial_j p} + \widetilde{w_j\partial_i p} \right] \tag{86}$$

$$\Xi^r_{ij} = \widetilde{y_i w_k \partial_k w_j} + \widetilde{w_j y_k \partial_k y_i} + \widetilde{p\partial_i w_j} + \widetilde{p\partial_j y_i}. \tag{87}$$

Finally, the viscous dissipation terms are:

$$\epsilon^y_{kikj} = -\nu\tau^\ell_{kikj}(\nabla\boldsymbol{y},\nabla\boldsymbol{y}) \tag{88}$$

$$\epsilon^w_{kikj} = -\nu\tau^\ell_{kikj}(\nabla\boldsymbol{w},\nabla\boldsymbol{w}) \tag{89}$$

$$\epsilon^r_{kikj} = -2\nu\tau^\ell_{kikj}(\nabla\boldsymbol{y},\nabla\boldsymbol{w}). \tag{90}$$

All of the above terms are written in their anisotropic form for completeness and to emphasize the peculiar non-linearity in IMHD. When using the isotropic form, the scale-to-scale transfer equations read:

$$\Pi^y_{ii} = -\tau^\ell_{ik}(\boldsymbol{y},\boldsymbol{w})\partial_k\widetilde{y}_i \tag{91}$$

$$\Pi^w_{ii} = -\tau^\ell_{ik}(\boldsymbol{w},\boldsymbol{y})\partial_k\widetilde{w}_i \tag{92}$$

$$\Pi^r_{ii} = -\tau^\ell_{ik}(\boldsymbol{y},\boldsymbol{w})\partial_k\widetilde{w}_i - \widetilde{y}_i\partial_k\widetilde{w}_k\widetilde{w}_i - \tau^\ell_{ik}(\boldsymbol{w},\boldsymbol{y})\partial_k\widetilde{y}_i - \widetilde{w}_i\partial_k\widetilde{y}_k\widetilde{y}_i. \tag{93}$$

Note that the pressure terms become surface terms using incompressibility, and thus are not scale-to-scale transfer terms. Similarly, the isotropic inter-scale transfers can be written as:

$$\Xi_{ij}^r = \widetilde{y_i w_k \partial_k w_i} + \widetilde{w_i y_k \partial_k y_i}\,. \tag{94}$$

Note here that the $y, w$ inter-scale and cross-field terms become surface terms. Finally, the viscous dissipation terms in their isotropic form are given by:

$$\epsilon_{kiki}^y = -\nu\tau_{kiki}^\ell(\nabla\mathbf{y}, \nabla\mathbf{y}) \tag{95}$$

$$\epsilon_{kiki}^w = -\nu\tau_{kiki}^\ell(\nabla\mathbf{w}, \nabla\mathbf{w}) \tag{96}$$

$$\epsilon_{kiki}^r = -2\nu\tau_{kiki}^\ell(\nabla\mathbf{y}, \nabla\mathbf{w})\,. \tag{97}$$

In this case, the $y, w$ viscous dissipation terms are positive definite, while this is not necessarily guaranteed for the cross-field viscous term.

## 5. Primitive Variables

Thus far, we work with the equations written in terms of the Elsasser variables, as they provide easier calculations. It is now possible to recover the energy and cross helicity versions in the primitive variables, by using the following equalities:

$$y_i y_j = (u_i + b_i)(u_j + b_j) = u_j u_j + u_i b_j + b_i u_j + b_i b_j$$
$$w_i w_j = (u_i - b_i)(u_j - b_j) = u_j u_j - u_i b_j - b_i u_j + b_i b_j$$
$$y_i w_j = (u_i + b_i)(u_j - b_j) = u_i u_j - u_i b_j + b_i u_j - b_i b_j\,,$$

Additionally, we introduce the following definitions of the large-scale energy, cross-field and cross helicity densities:

$$\widetilde{e}_{ij}^E = \frac{1}{2}\left(\widetilde{u}_i\widetilde{u}_j + \widetilde{b}_i\widetilde{b}_j\right) = \frac{1}{4}\left(\widetilde{y}_i\widetilde{y}_j + \widetilde{w}_i\widetilde{w}_j\right) \tag{98}$$

$$\widetilde{e}_{ij}^H = \frac{1}{2}\left(\widetilde{u}_i\widetilde{b}_j + \widetilde{b}_i\widetilde{u}_j\right) = \frac{1}{4}\left(\widetilde{y}_i\widetilde{y}_j - \widetilde{w}_i\widetilde{w}_j\right) \tag{99}$$

$$\widetilde{e}_{ij}^r = \widetilde{u}_i\widetilde{u}_j - \widetilde{u}_i\widetilde{b}_j + \widetilde{b}_i\widetilde{u}_j - \widetilde{b}_i\widetilde{b}_j\,, \tag{100}$$

as well as the small-scale densities:

$$\bar{e}_{ij}^E = \frac{1}{2}\left(\tau_{ij}(\mathbf{u}, \mathbf{u}) + \tau_{ij}(\mathbf{b}, \mathbf{b})\right) = \frac{1}{4}\left(\tau_{ij}(\mathbf{y}, \mathbf{y}) + \tau_{ij}(\mathbf{w}, \mathbf{w})\right) \tag{101}$$

$$\bar{e}_{ij}^H = \frac{1}{2}\left(\tau_{ij}(\mathbf{u}, \mathbf{b}) + \tau_{ij}(\mathbf{b}, \mathbf{u})\right) = \frac{1}{4}\left(\tau_{ij}(\mathbf{y}, \mathbf{y}) - \tau_{ij}(\mathbf{w}, \mathbf{w})\right) \tag{102}$$

$$\bar{e}_{ij}^r = \tau_{ij}(\mathbf{u}, \mathbf{u}) - \tau_{ij}(\mathbf{u}, \mathbf{b}) + \tau_{ij}(\mathbf{b}, \mathbf{u}) - \tau_{ij}(\mathbf{b}, \mathbf{b}) = \tau_{ij}(\mathbf{y}, \mathbf{w})\,, \tag{103}$$

and derive the time-evolution equations for all of the above terms.

### *5.1. Transfer Terms*

The energy and cross-helicity are easily recovered by adding and subtracting the transfer terms from before:

$$\Pi_{ij}^E = \frac{1}{2}\left[(\tau_{jk}^\ell(\boldsymbol{u},\boldsymbol{u}) - \tau_{jk}^\ell(\boldsymbol{b},\boldsymbol{b}))\partial_k\widetilde{u}_i + (\tau_{ik}^\ell(\boldsymbol{u},\boldsymbol{u}) - \tau_{ik}^\ell(\boldsymbol{b},\boldsymbol{b}))\partial_k\widetilde{u}_j\right.$$
$$\left. + (\tau_{jk}^\ell(\boldsymbol{b},\boldsymbol{u}) - \tau_{jk}^\ell(\boldsymbol{u},\boldsymbol{b}))\partial_k\widetilde{b}_i + (\tau_{ik}^\ell(\boldsymbol{b},\boldsymbol{u}) - \tau_{ik}^\ell(\boldsymbol{u},\boldsymbol{b}))\partial_k\widetilde{b}_j + (\widetilde{u}_i\partial_j\widetilde{p} + \widetilde{u}_j\partial_i\widetilde{p})\right] \tag{104}$$

$$\Pi_{ij}^H = \frac{1}{2}\left[(\tau_{jk}^\ell(\boldsymbol{b},\boldsymbol{b}) - \tau_{jk}^\ell(\boldsymbol{u},\boldsymbol{u}))\partial_k\widetilde{b}_i + (\tau_{ik}^\ell(\boldsymbol{b},\boldsymbol{b}) - \tau_{ik}^\ell(\boldsymbol{u},\boldsymbol{u}))\partial_k\widetilde{b}_j\right.$$
$$\left. + (\tau_{jk}^\ell(\boldsymbol{u},\boldsymbol{b}) - \tau_{jk}^\ell(\boldsymbol{b},\boldsymbol{u}))\partial_k\widetilde{u}_i + (\tau_{ik}^\ell(\boldsymbol{u},\boldsymbol{b}) - \tau_{ik}^\ell(\boldsymbol{b},\boldsymbol{u}))\partial_k\widetilde{u}_j + (\widetilde{b}_i\partial_j + \widetilde{b}_j\partial_i)\widetilde{p}\right]. \tag{105}$$

Note that we do not include the cross-field term here, as it diverges. As for the inter-scale terms:

$$\Xi_{ij}^E = -\frac{1}{2}\left[\widetilde{u_i\partial_j p} + \widetilde{u_j\partial_i p}\right] \tag{106}$$

$$\Xi_{ij}^H = -\frac{1}{2}\left[\widetilde{b_i\partial_j p} + \widetilde{b_j\partial_i p}\right]. \tag{107}$$

Again, it is not advantageous to write out the cross-field term. For the viscous dissipation terms:

$$\epsilon_{kikj}^E = -\nu\left[\tau_{kikj}^\ell(\nabla\boldsymbol{u},\nabla\boldsymbol{u}) + \tau_{kikj}^\ell(\nabla\boldsymbol{b},\nabla\boldsymbol{b})\right] \tag{108}$$

$$\epsilon_{kikj}^H = -\nu\left[\tau_{kikj}^\ell(\nabla\boldsymbol{u},\nabla\boldsymbol{b}) + \tau_{kikj}^\ell(\nabla\boldsymbol{b},\nabla\boldsymbol{u})\right] \tag{109}$$

$$\epsilon_{kikj}^{re} = -2\nu\left[\tau_{kikj}^\ell(\nabla\boldsymbol{u},\nabla\boldsymbol{u}) - \tau_{kikj}^\ell(\nabla\boldsymbol{u},\nabla\boldsymbol{b}) + \tau_{kikj}^\ell(\nabla\boldsymbol{b},\nabla\boldsymbol{u}) - \tau_{kikj}^\ell(\nabla\boldsymbol{b},\nabla\boldsymbol{b})\right]. \tag{110}$$

The isotropic versions can also be derived:

$$\Pi_{ii}^E = (\tau_{ik}^\ell(\boldsymbol{u},\boldsymbol{u}) - \tau_{ik}^\ell(\boldsymbol{b},\boldsymbol{b}))\partial_k\widetilde{u}_i + (\tau_{ik}^\ell(\boldsymbol{b},\boldsymbol{u}) - \tau_{ik}^\ell(\boldsymbol{u},\boldsymbol{b}))\partial_k\widetilde{b}_i$$
$$\Pi_{ii}^H = (\tau_{ik}^\ell(\boldsymbol{b},\boldsymbol{b}) - \tau_{ik}^\ell(\boldsymbol{u},\boldsymbol{u}))\partial_k\widetilde{b}_i + (\tau_{ik}^\ell(\boldsymbol{u},\boldsymbol{b}) - \tau_{ik}^\ell(\boldsymbol{b},\boldsymbol{u}))\partial_k\widetilde{u}_i. \tag{111}$$

Again, the inter-scale terms become surface flux terms, and the isotropic viscous dissipation terms read:

$$\epsilon_{kiki}^E = -\nu\left[\tau_{kiki}^\ell(\nabla\boldsymbol{u},\nabla\boldsymbol{u}) + \tau_{kiki}^\ell(\nabla\boldsymbol{b},\nabla\boldsymbol{b})\right] \tag{112}$$

$$\epsilon_{kiki}^H = -2\nu\tau_{kiki}^\ell(\nabla\boldsymbol{u},\nabla\boldsymbol{b}) \tag{113}$$

$$\epsilon_{kikj}^{re} = -2\nu\left[\tau_{kiki}^\ell(\nabla\boldsymbol{u},\nabla\boldsymbol{u}) - \tau_{kiki}^\ell(\nabla\boldsymbol{b},\nabla\boldsymbol{b})\right]. \tag{114}$$

## 6. Estimating the Scale-to-Scale Energy Transfer

In this section, we show how to estimate the behaviour of the above terms making use of the Hölder inequality and of Lebesque space norms ($L_p$-norms). This naturally opens the path towards Kolmogorov-like phenomenologies that incorporate intermittency and other phenomena. Here, we only use the isotropic forms of $y$, $w$-fields and the energy/helicity, as those are the ideal invariants of IMHD. In this section, we work out the estimate on the $y$-field scale-to-scale transfer, and generalize to the others.

### 6.1. Structure Functions

Some simple identities on the cumulants can be proved by noting that $\int_{V(\ell)} d^3x'\, \Im_\ell(\boldsymbol{r})f(\boldsymbol{x}) = f(\boldsymbol{x})\int_{V(\ell)} d^3x'\, \Im_\ell(\boldsymbol{r}) = f(\boldsymbol{x})$, so that we can write:

$$\tau_{ij}^\ell(f,g) = \tau_{ij}^\ell(\delta f,\delta g) \tag{115}$$

$$\tau_{ijk}^\ell(f,g,h) = \tau_{ij}^\ell(\delta f,\delta g,\delta h). \tag{116}$$

Additionally, by noting that $\int_{V(\ell)} d^3 x' \, r_j/|r_i| \, \partial_U \Im_\ell(U) = 0$; $U = |r_j|/\ell$, we can write:

$$\partial_j \tilde{f} = \int_{V(\ell)} d^3 x' \, \ell^{-1} \frac{r_j}{|r_i|} \partial_U \Im_\ell(U) \, \delta f \,, \tag{117}$$

where we made use of previous definition of the increment $\delta f = f(x') - (x)$.

### 6.2. Estimating the Scale-to-Scale Energy Flux

To estimate the scale-to-scale energy flux in the $y$-field term, we can write it as:

$$\Pi_{ii}^y = -\tau_{ij}^\ell(\delta y, \delta w) \partial_j \tilde{y}_i \,. \tag{118}$$

We start the derivation by making the following general statement, following the definition of the $L_p$-norm:

$$||\tilde{f}||_p = \left[ \int_V d^3 x \left| \int_{V(\ell)} d^3 x' \, \Im_\ell(x - x') f(x') \right|^p \right]^{1/p} . \tag{119}$$

Then, applying Young's convolutions theorem:

$$||\tilde{f}||_p \leq ||\Im_\ell(x - x')||_q \, ||f(x')||_r \,; \quad 1 + \frac{1}{p} = \frac{1}{q} + \frac{1}{r} \,, \tag{120}$$

where we assume that these quantities are integrable over both $d^3 x$ and $d^3 x'$. Choosing $q = 1$ and noting that the coarse-grain kernel is positive defined and normalized, we recover:

$$||\tilde{f}||_p \leq ||f(x')||_r \,; \quad p = r \,, \tag{121}$$

a very useful result for this approach. A similar estimate can be worked out for the second-order cumulant of the energy transfer term of interest, first by means of the Minkowski inequality:

$$\begin{aligned}
||\tau_{ij}^\ell(\delta w, \delta y)||_p &\leq \left\| \int_{V(\ell)} d^3 x' \, \Im_\ell \delta w_i \delta y_j \right\|_p + \left\| \int_{V(\ell)} d^3 x' \, \Im_\ell \delta w_i \int_{V(\ell)} d^3 x' \, \Im_\ell \delta y_j \right\|_p \\
&\leq \left\| \int_{V(\ell)} d^3 x' \, \Im_\ell \delta w_i \delta y_j \right\|_p + \left\| \int_{V(\ell)} d^3 x' \, \Im_\ell \delta w_i \right\|_{p_w} \left\| \int_{V(\ell)} d^3 x' \, \Im_\ell \delta y_j \right\|_{p_y} .
\end{aligned} \tag{122}$$

In the second line of the above equation, we use the Hölder inequality so that $1/p = 1/p_w + 1/p_y$. Using then Young's inequality:

$$||\tau_{ij}^\ell(\delta w, \delta y)||_p \leq ||\delta w_i \delta y_j||_p + ||\delta w_i||_{p_w} ||\delta y_j||_{p_y} \,. $$

Using again the Hölder inequality leads to the following relation:

$$||\tau_{ij}^\ell(\delta w, \delta y)||_p \leq ||\delta w_i||_{p'_w} ||\delta y_j||_{p'_y} + ||\delta w_i||_{p_w} ||\delta y_j||_{p_y} \,, \tag{123}$$

where we have $1/p = 1/p'_w + 1/p'_y$ and in general $p'_\alpha \neq p_\alpha$. As a next step, we want to provide an estimate for the derivative $\partial_j \tilde{w}_i$, which can be written as follows with the increment:

$$\partial_j \tilde{w}_i = \int_{V(\ell)} d^3 x' \, \ell^{-1} \frac{r_j}{|r_i|} \partial_U \Im_\ell(U) \, \delta w_i \,. \tag{124}$$

With this form, the Young inequality reads:

$$\left|\left|\partial_j \widetilde{w}_i\right|\right|_q \leq \ell^{-1} \left|\left|\frac{r_j}{|r_i|}\delta w_i\right|\right|_{q_1} ||\partial_U \mathfrak{S}_\ell(U)||_{q_2}^{1}, \tag{125}$$

which is valid for an infinitely differentiable coarse-grain kernel and for any choice of $q_2$. By setting $q_2 = \infty \Rightarrow q = q_1$, and using again the Hölder inequaltiy, we obtain:

$$\left|\left|\partial_j \widetilde{w}_i\right|\right|_q \leq \ell^{-1} \left|\left|\frac{r_j}{|r_i|}\right|\right|_{q_3} ||\delta w_i||_{q_4}. \tag{126}$$

The first norm in the right-hand side of the above equation is a non-important number for $q_3$, thus, choosing $q_3 = \infty$, we have:

$$\left|\left|\partial_j \widetilde{w}_i\right|\right|_q \leq C' \ell^{-1} ||\delta w_i||_q. \tag{127}$$

Putting all these estimates together, and using the Hölder inequality once more, we have:

$$\begin{aligned}
\left|\left|\tau_{ij}^\ell(\delta\boldsymbol{w},\delta\boldsymbol{y})\,\partial_j\widetilde{w}_i\right|\right|_r &\leq \left|\left|\tau_{ij}^\ell(\delta\boldsymbol{w},\delta\boldsymbol{y})\right|\right|_p \left|\left|\partial_j\widetilde{w}_i\right|\right|_q \\
&\leq \ell^{-1}||\delta w_i||_q ||\delta w_i||_{p'_w} ||\delta y_j||_{p'_y} + \ell^{-1}||\delta w_i||_q ||\delta w_i||_{p_w} ||\delta y_j||_{p_y}.
\end{aligned} \tag{128}$$

If we consider the increments to be Besov regular, the change of variables is permitted and we write the increment as:

$$||\delta f_i||_p \leq C_p |r_i|^{\alpha_p}; \quad 0 < \alpha < 1, \tag{129}$$

we can finally write:

$$\left|\left|\tau_{ij}^\ell(\delta\boldsymbol{w},\delta\boldsymbol{y})\,\partial_j\widetilde{w}_i\right|\right|_r \leq \ell^{-1} C_{q,p'_w,p'_y} |r_i|^{\alpha_{q,p'_w,p'_y}} + \ell^{-1} C_{q,p_w,p_y} |r_i|^{\alpha_{q,p_w,p_y}}, \tag{130}$$

where the following definitions have been used:

$$C_{q,p'_w,p'_y} = C_q C_{p'_w} C_{p'_y} \tag{131}$$

$$C_{q,p_w,p_y} = C_q C_{p_w} C_{p_y} \tag{132}$$

$$\alpha_{q,p'_w,p'_y} = \alpha_q + \alpha_{p'_w} + \alpha_{p'_y} \tag{133}$$

$$\alpha_{q,p_w,p_y} = \alpha_q + \alpha_{p_w} + \alpha_{p_y}. \tag{134}$$

*6.3. Onsager's "Minimal Assumption" for Incompressible Magnetohydrodynamics*

The idea of defining the minimal conditions necessary to obtain a turbulent cascade is well-founded in hydrodynamics, but it takes a different enlightenment in IMHD. Consider the energy flux term:

$$\left|\left|\tau_{ij}^\ell(\delta\boldsymbol{w},\delta\boldsymbol{y})\,\partial_j\widetilde{w}_i\right|\right|_r \leq C_{q,p'_w,p'_y} |\ell|^{\alpha_{q,p'_w,p'_y}-1} + C_{q,p_w,p_y} |\ell|^{\alpha_{q,p_w,p_y}-1}. \tag{135}$$

For the real fields ($\ell \to 0$), the energy transfer rate goes to zero, unless the following condition is satisfied:

$$\alpha_{q,p'_w,p'_y} \leq 1, \quad \alpha_{q,p_w,p_y} \leq 1. \tag{136}$$

In hydrodynamics, we have $w_j = y_i$, as is usually reported, leading to the standard result:

$$\alpha_q = \alpha_{p'_w} = \alpha_{p'_y} \Rightarrow \alpha_q \leq 1/3. \tag{137}$$

When the equality is taken, the Kolmogorov's 1941 scaling is retrieved, for which homogeneity is necessary. For real turbulence, there is a cut-off at the Kolmogorov scale $\eta$. However, in the case of IMHD, the exponents are certainly not equivalent because there are two fields involved, thus the inequality

$$\alpha_{q,p_w,p_y} = \alpha_q + \alpha_{p_w} + \alpha_{p_y} \leq 1 \tag{138}$$

can be satisfied without the restrictions of hydrodynamics (Equation (137)). Namely, one of the two fields can be *smooth* while the other is free to be *irregular*, and the cascade will still occur. This can be seen in Equation (138) as the sum of the three exponents (corresponding to two fields, either the Elsasser variables or the primitive variables) must satisfy the inequality giving freedom to the exponents, not permitted in the hydrodynamic case.

*6.4. Scalings for Homogenous IMHD*

For the inertial range of statistically homogeneous turbulence, the scaling of the energy flux terms must be linear [12]. This argument does require consideration of geometry, but we only use it here to motivate the following discussion. For the energy transfer, this takes on the form:

$$|\Pi_{ii}^E| \leq |\tau_{ik}^\ell(\boldsymbol{u},\boldsymbol{u})\partial_k \widetilde{u}_i| + |\tau_{ik}^\ell(\boldsymbol{b},\boldsymbol{b})\partial_k \widetilde{u}_i| + |\tau_{ik}^\ell(\boldsymbol{b},\boldsymbol{u})\partial_k \widetilde{b}_i| + |\tau_{ik}^\ell(\boldsymbol{u},\boldsymbol{b})\partial_k \widetilde{b}_i| \tag{139}$$

following Section 6 to give the scaling:

$$|\Pi_{ii}^E| \leq |r_i|^{\alpha_{uuu}-1} + |r_i|^{\alpha_{bbu}-1} + |r_i|^{\alpha_{bub}-1} + |r_i|^{\alpha_{ubb}-1}. \tag{140}$$

The subscripts of the $\alpha$s denote the terms estimated as in Equation (138). The principal of scale independent energy transfer (for example, $\alpha_{bbu} = 1$ ) is the main result of the article. Here, we note that, for multi-fractal fields, the global scaling relations appear:

$$\langle f(|r_i|)^p \rangle \sim |r_i|^{\gamma(p)} \tag{141}$$

where $p = 2$ is the second-order structure function, our primary data analysis object, and the ratios $\gamma(p)/\gamma(p')$ are not obvious, also known as intermittency. Our result (Equation (140)), for one of the exponents in terms of multi-fractal fields, takes on the form:

$$\langle \delta b_i^p \rangle \sim |r_i|^{\gamma_b(p)} \quad \Rightarrow \alpha_{bub} = 2\gamma_b(p=1) + \gamma_u(p=1) = 1 \tag{142}$$

This is important to point out, because our relation necessitates the functions $\gamma_b(p)$ to be known to predict the power spectrum.

## 7. Application to Observation

We finish by demonstrating the significance of the calculations described in this paper to the structure functions measured in space plasmas, as for example in the solar wind, as discussed in the Introduction. In Section 6, we present a general method for estimating the scale-to-scale energy transfers using the notation of an upper bound. The use of indicator functions and some differentiability conditions for the fields lead us to conclude a reasonable bound on the second-order cumulants to be:

$$\left|\tau_{ij}^\ell(\delta\boldsymbol{w},\delta\boldsymbol{y})\right| \leq \left|\delta w_i \delta y_j\right| \Rightarrow \left|\delta w_i \delta y_j\right| \leq C_{y,w}|r_i|^{\alpha_{y,w}}, \tag{143}$$

retaining the correlation between the two field increments. This suggests that the scalings of these functions follow a balance relation. Recalling the form of the scale-to-scale energy transfer:

$$\Pi_{ii}^E = \left(\tau_{ik}^{\ell}(\boldsymbol{u},\boldsymbol{u}) - \tau_{ik}^{\ell}(\boldsymbol{b},\boldsymbol{b})\right)\partial_k \widetilde{u}_i + \left(\tau_{ik}^{\ell}(\boldsymbol{b},\boldsymbol{u}) - \tau_{ik}^{\ell}(\boldsymbol{u},\boldsymbol{b})\right)\partial_k \widetilde{b}_i, \tag{144}$$

some straightforward steps lead us to suggest the following balance relations:

$$|\Pi_{ii}^E| \leq |r_i|^{\alpha_{E_r,u}-1} + |r_i|^{\alpha_{E_c,b}-1}, \tag{145}$$

where the following definitions have been used:

$$\alpha_{E_r,u} = \alpha_{E_r} + \alpha_u, \ \alpha_{E_c,b} = \alpha_{E_c} + \alpha_b \tag{146}$$

$$\left|\left(\tau_{ik}^{\ell}(\delta\boldsymbol{u},\delta\boldsymbol{u}) - \tau_{ik}^{\ell}(\delta\boldsymbol{b},\delta\boldsymbol{b})\right)\right| \leq C_{E_r}|r_i|^{\alpha_{E_r}}, \ \left|\left(\tau_{ik}^{\ell}(\delta\boldsymbol{b},\delta\boldsymbol{u}) - \tau_{ik}^{\ell}(\delta\boldsymbol{u},\delta\boldsymbol{b})\right)\right| \leq C_{E_c}|r_i|^{\alpha_{E_c}}. \tag{147}$$

This retains the correlations of the fields (e.g., $\tau_{ik}^{\ell}(\delta\boldsymbol{b},\delta\boldsymbol{b})$) into the scaling of the scale-to-scale energy transfer estimate. This is the unique result of this article. We obtain scaling relations that involve the mixed second-order cumulants. This gives a better understanding of the role that the cross-helicity and residual energy play in IMHD turbulence.

## 7.1. Analysis

The IMHD equations are often invoked as a model for the large spatiotemporal scales of the solar wind. We test the relations (Equation (145)) directly using data from the WIND spacecraft [16,17]. In 2004, the WIND spacecraft settled to the Lagrange point between the Earth and Sun where it made measurements of the pristine solar wind. This provides a public dataset at $\sim$3 s cadence of the magnetic field and velocity field. We subset the year 2005 into 6 h intervals and organize the data by the quantities,

$$\sigma_r = \frac{y_i' w_i'}{y_i' y_i' + w_i' w_i'}, \ \sigma_c = \frac{v_i' b_i'}{v_i' v_i' + b_i' b_i'} \tag{148}$$

normalized cross helicity $\sigma_c$ and the normalized residual energy $\sigma_r$ defined by fluctuating quantities (i.e., $u_i' = u_i - \langle u_i \rangle_T$) of the velocity field $v_i$, Alfvèn velocity $b_i$ and the Elsasser variables $y_i = u_i + b_i$, $w_i = u_i - b_i$. We use 6 h $= T$ for the Reynold's decomposed fluctuations. This restricts the data ($\sigma_r^2 + \sigma_c^2 \leq 1$) due to the geometry of the vectors [18]. In Figure 1, we provide a two-dimensional histogram of the 1158 6-h intervals conditioned on $(\sigma_c, \sigma_r)$. We then take each individual 6-h interval and measure the structure functions:

$$S_r(\tau) = \langle (y_i(t+\tau) - y_i(t))(w_i(t+\tau) - w_i(t)) \rangle_T \tag{149}$$
$$S_c(\tau) = \langle (u_i(t+\tau) - u_i(t))(b_i(t+\tau) - b_i(t)) \rangle_T \tag{150}$$
$$S_u(\tau) = \langle (u_i(t+\tau) - u_i(t)) \rangle_T \tag{151}$$
$$S_b(\tau) = \langle (b_i(t+\tau) - b_i(t)) \rangle_T \tag{152}$$

The lag $\tau$ is studied for the corresponding inertial range of IMHD and the interval time span ($T$) is six hours so that we cover many correlation times of the turbulence. We then fit the inertial range with a power law fit. In Figure 2, we condition the data again by ($\sigma_c, \sigma_r$) and show bins that are populated by more than five events. The color corresponds to the exponent of the power law fit.

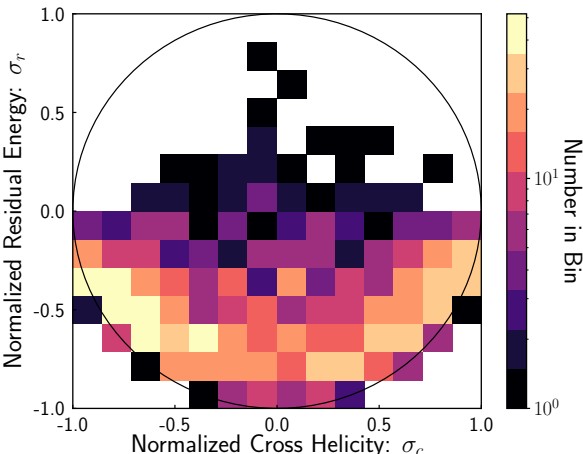

**Figure 1.** Two-dimensional histogram of the samples conditioned on the normalized cross helicity and normalized residual energy.

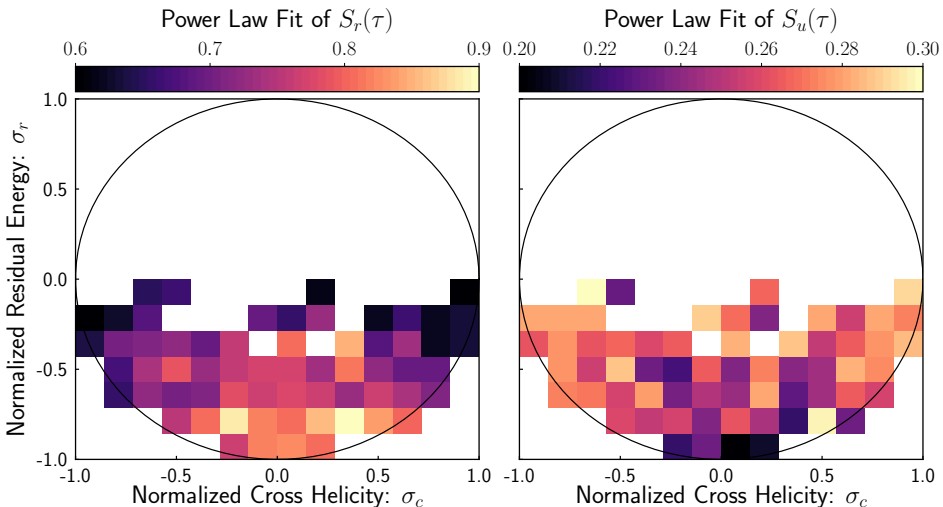

**Figure 2.** Bins, conditioned on the normalized cross helicity and normalized residual energy show the exponents of power law fits to structure functions. The structure functions correspond to definitions in the text.

### 7.2. Discussion of Analysis

We tested the hypotheses of Section 7 by measuring the exponents of power law fits to structure functions. Figure 2 displays the steepening (i.e., larger exponent) for lighter colors, which, between the two plots, the structure functions steepen in different regions, balancing each other. This is the expectation considering statistical homogeneity, that of a scale independent energy transfer rate. Having the correct intermittency model, as pointed out in Section 6.4, would better serve the exact relations. To the knowledge of the authors, these results are new.

### 8. Conclusions

In this article, we propose a step-by-step illustration of the use of the filtered approach to obtain the equations describing the scale-by-scale energy transfer in incompressible magnetohydrodynamic turbulence. While most of the results presented here are not new, we believe a detailed description of the calculation can contribute to clarify the procedure, and help in the possible extension to other systems, such as Hall MHD. Additionally, we show how this approach naturally leads to the estimation of the Kolmogorov spectrum, without the need of invoking phenomenological arguments. We have identified a balance relation between the scaling of the energy transfer associated with the different

components, which could be verified using numerical simulations, and eventually be of interest for application to space plasmas and in particular for the determination of the different contributions to the global energy transfer. Finally, we used solar wind data from the WIND spacecraft to measure the exponents of the structure functions, alluded to in Section 7, and found the balance relations being satisfied. Any lack of perfect agreement can be attributed to the insufficient sampling for statistical homogeneity and the use of a mono-fractal model. We leave this for future work. The balance relations give a better understanding of the role of the normalized residual energy and normalized cross helicity in IMHD turbulence.

**Author Contributions:** J.T.C. contributed the primary text and analysis. L.S.-V. supervised text and analysis.

**Funding:** J.T.C. is currently a graduate student at Queen Mary University of London where he is supported by a Queen Mary Principal Studentship. This research was completed on independent time.

**Conflicts of Interest:** The authors declare no conflict of interest.

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
