# Peer review of "Energy Transfer in Incompressible Magnetohydrodynamics: The Filtered Approach"

_fluids, doi:10.3390/fluids4030163_

Round 1

Reviewer 1 Report

Referee report : In this paper, the authors developed an incompressible magnetohydrodynamic (IMHD) energy budget equations with a spatial filtering kernel. These equations are used to estimate the scaling of the structure functions. The authors test these new equations in the Politano-Pouquet law limit. This work can help to reproduce Kolmogorov spectrum, without the need of invoking phenomenological arguments. I can recommend the paper for publication after following improvements. Section 6 At 6.3, in paragraphe 3 change In the lignes 103-104 : Could authors explain more how one of the fields can smooth whilte the other is irregular and the cascade still occur and the link after wit Kolmogorov cascade. ‘the the Kolmogorov’ ->’ the Kolmogorov’ Application to Observations Could authors add in this section some quantitative study in the solar wind case.

Author Response

We thank the referee for reviewing the manuscript. The grammar error has been fixed. The text includes a better description (where the referee asked) and we have provided an analysis.

Reviewer 2 Report

This paper discusses the energy transfer and dissipation in incompressible MHD. The authors derive the energy transfer equations and eventually obtain the Ideal IMHD equations and eventually estimate the scale-to-scale energy transfer. In the last section before conclusions they discuss applications to observations and finally they conclude. 

- An important weakness of the paper is the connection to the observations as there is no such connection evident by the text that is there. In particular there is no comparison of their results to actual observations, even in a qualitative level. This needs to be improved. 

- The overall discussion is quite descriptive, and does not illustrate much physical insight, namely the equations are derived but not discussed. It feels that a large mathematical derivation takes place, but the actual discussion is limited. 

- The literature review is limited and has not incorporate some interesting recent results. I would suggest to enhance their literature review (i.e. they may consider discussing Mason et al. Phys.Rev.E77:036403,2008; Galtier & Buchlin Astrophys.J. 656 (2007) 560-566). 

Author Response

We thank the referee for the suggestions and have gone ahead and provided an analysis that supports some of the main results in the text.

We have decided to not incorporate the references. Our approach is different than most modern work and we point out that the -3/2 versus -5/3 debate is probably covered by our analysis, as we do not make assumptions (e.g. balanced or imbalanced turbulence) and report the balance relations, which we believe to be more fundamental. It is future work to check if indeed this debate is better understood via the balance relations written here.